# DyCC-Net: Dynamic Context Collection Network for Input-Aware Drone-View Object Detection

**Yue Xi** [1], **Wenjing Jia** [2], **Qiguang Miao** [3,*], **Xiangzeng Liu** [3], **Xiaochen Fan** [4] and **Jian Lou** [1]

1 Guangzhou Institute of Technology, Xidian University, Guangzhou 510555, China
2 Global Big Data Technologies Centre, University of Technology Sydney, Ultimo, NSW 2007, Australia
3 School of Computer Science and Technology, Xidian University, Xi'an 710071, China
4 Department of Electronic Engineering, Tsinghua University, Beijing 100084, China
* Correspondence: qgmiao@xidian.edu.cn

**Abstract:** Benefiting from the advancement of deep neural networks (DNNs), detecting objects from drone-view images has achieved great success in recent years. It is a very challenging task to deploy such DNN-based detectors on drones in real-life applications due to their excessive computational costs and limited onboard computational resources. Large redundant computation exists because existing drone-view detectors infer all inputs with nearly identical computation. Detectors with less complexity can be sufficient for a large portion of inputs, which contain a small number of sparse distributed large-size objects. Therefore, a drone-view detector supporting input-aware inference, i.e., capable of dynamically adapting its architecture to different inputs, is highly desirable. In this work, we present a **Dy**namic **C**ontext **C**ollection **Net**work (DyCC-Net), which can perform input-aware inference by dynamically adapting its structure to inputs of different levels of complexities. DyCC-Net can significantly improve inference efficiency by skipping or executing a context collector conditioned on the complexity of the input images. Furthermore, since the weakly supervised learning strategy for computational resource allocation lacks of supervision, models may execute the computationally-expensive context collector even for easy images to minimize the detection loss. We present a **Pseudo**-label-based semi-supervised **Learning** strategy (Pseudo Learning), which uses automatically generated pseudo labels as supervision signals, to determine whether to perform context collector according to the input. Extensive experiment results on VisDrone2021 and UAVDT, show that our DyCC-Net can detect objects in drone-captured images efficiently. The proposed DyCC-Net reduces the inference time of state-of-the-art (SOTA) drone-view detectors by over 30 percent, and DyCC-Net outperforms them by 1.94% in $AP_{75}$.

**Keywords:** object detection in drone-view images; dynamic neural network; pseudo-label learning

## 1. Introduction

Recently, Unmanned Aerial Vehicles (UAVs), commonly known as drones, have attracted much attention [1,2]. Drones can be deployed rapidly at a relatively low cost, in various emerging applications, e.g., aerial photography and video surveillance [3,4]. Intelligent processing of images or videos captured by drones is very demanding, which combines the advancements in computer vision and drones closely. Taking advantage of advances in Deep Neural Networks (DNNs) in remote sensing image processing, remarkable advances have been achieved in drone-view object detection, which aims to detect instances of objects from images captured by drones.

Existing methods focus on extracting robust features to distinguish foreground targets which contain very limited number of pixels, from the background clutter [5–7]. There are three main types of drone-view detectors: context-based methods [8–10], super-resolution-based (SR-based) methods [11–13], and multi-scale representation-based (MR-based) methods [14–16]. Despite their success in detecting objects from images captured by drones, the deployment of these detectors on drones in the real-world can be challenging. A major

reason is the conflict between these models' high computational costs and the very limited onboard computational resources. Existing drone-view detectors typically consist of complicated modules for super-resolution or context collection of Regions of Interest (RoIs) to achieve maximum accuracy. They typically involve prohibitively high computational costs.

We argue that there are large redundant computations because the existing drone-view detectors [12,17,18] perform inference in a static manner. They infer all inputs with a fixed computational graph, and, thus, can not adapt to the varying complexity of the input during inference. Figure 1 shows an example where statically and dynamically configured detectors infer easy and hard inputs by using a module for context collection. In many real-life scenes, only a small portion of inputs require the module to be specially designed for super-resolving RoIs or encoding context information. Consequently, it becomes highly desirable to design a drone-view detector with a dynamic architecture, which improves its computational efficiency through input-aware inference.

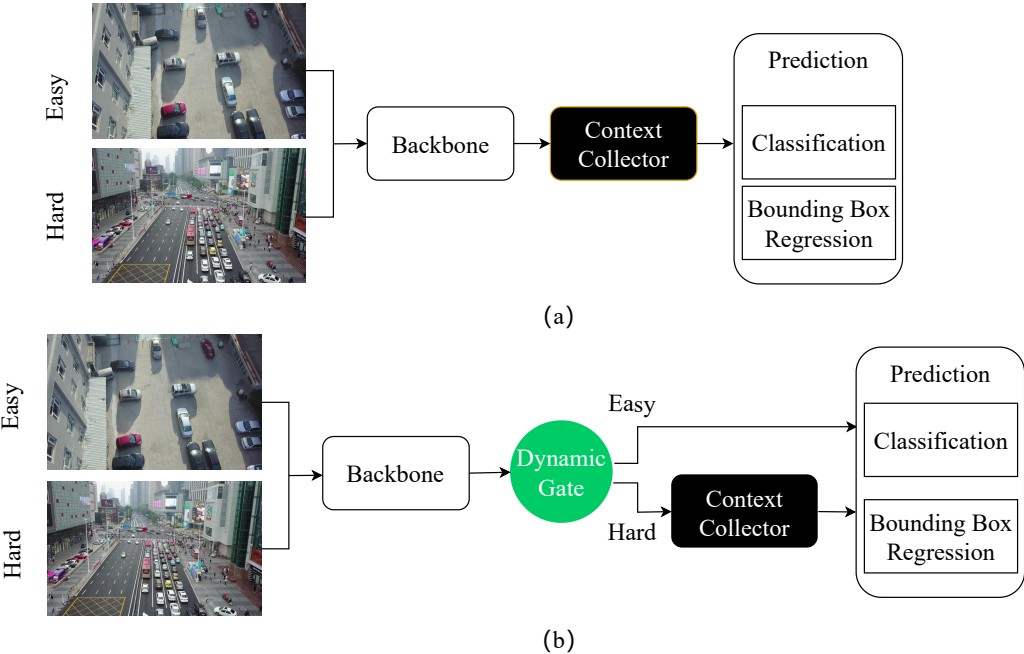

**Figure 1.** Easy (sparsely distributed and large-size) and Hard (crowded and small-size) inputs for detectors in static and dynamic cases. (**a**) In the static case, a detector executes the operation of context collection to infer inputs regardless of their complexity. (**b**) In the dynamic case, a dynamic gate is introduced to select an appropriate path for different inputs. The detector with the gate module can skip context collection for easy inputs, and only execute context collection for hard inputs.

In this work, we present **Dy**namic **C**ontext **C**ollection **Net**work (DyCC-Net), a drone-view detector supporting input-aware inference. DyCC-Net can perform input-aware inference to offer a balanced trade-off between computational costs and accuracy. It can skip or execute a Context Collector module [19] during inference, depending on the complexity of the inputs. Specifically, DyCC-Net can skip the Context Collector module to reduce the computational cost for easy inputs that can be correctly recognized without context information. Meanwhile, it can obtain high accuracy by effectively recognizing small objects in hard inputs by executing context collection. Furthermore, since the weakly supervised learning strategy for computational resource allocation lacks supervision, training the model with detection loss only may cause that it selects the context collector even for easy images. We present a **Pseudo**-label-based semi-supervised **Learning** strategy "Pseudo Learning", which uses the generated pseudo labels as supervised signals to allocate appropriate computation resources effectively, depending on the inputs.

The key contributions can be summarized as follows:

(1)  We present a drone-view detector supporting input-aware inference, called "DyCC-Net", which skips or executes a Context Collector module depending on inputs' complexity. Thus, it improves the inference efficiency by minimizing unnecessary computation. To the best of our knowledge, this work is the first study exploring dynamic neural networks on a drone-view detector.

(2)  We design a core dynamic context collector module and adopt the Gumbel–Softmax function to address the issue of training networks with discrete variables.

(3)  We propose a pseudo-labelling-based semi-supervised learning strategy, called "Pseudo Learning", which guides the process of allocating appropriate computation resources on diverse inputs, to achieve the speed-accuracy trade-off.

We evaluate our DyCC-Net on two widely used public datasets for drone-view object detection, i.e., VisDrone2021 and UAVDT. We compare DyCC-Net with 10 state-of-the-art (SOTA) methods and find that the proposed DyCC-Net reduces the inference time of SOTA models by over 30 percent. In addition, we also find that DyCC-Net outperforms the previous models by over 1.94% in $AP_{75}$.

The rest of the paper is organized as follows: First, we review the current development on drone-view object detection and give a summary of related works in Section 2. Section 3 introduces the preliminaries of DyCC-Net, including Feature Pyramid Network and Context Collector. Section 4 gives details of our DyCC-Net. Section 5 shows experiment results of DyCC-Net. Finally, the paper concludes in Section 6.

## 2. Related Work

DyCC-Net is related to two types of approaches, i.e., dynamic neural networks and drone-view object detection, which related works are reviewed in this section.

### 2.1. Dynamic Neural Networks

The research on Dynamic Neural Networks (DyNNs) is an emerging topic in deep learning academic community [20]. The parameters or structures of DyNNs can be adapted to different inputs [21]. They can make data-dependent decisions, adaptively determining whether a module should be skipped or executed conditioned on input data, to improve computational efficiency.

A variety of methods [21–25] focus on the design of dynamic architectures, which are adjusted based on each input during inference. There mainly exist two types of methods: dynamic depth and dynamic width. They allocate computational resources conditioned on the inputs. This has a great potential to reduce redundant computational costs. Dynamic-depth-based DyNNs attempted to adjust the network's depth conditioned on each input. Bolukbasi et al. [26] provided early exits in shallow layers, to reduce computational costs of executing deep layers for easy input based on the decision of a classifier. SkipNet [27] was proposed to enable dynamic layer skipping, which was a more flexible layer skipping paradigm. Dynamic-width-based DyNNs attempted to skip branches in Mixture-of-Experts (MoEs) to improve inference efficiency. Conventional soft MoEs [28,29] calculated soft weights to each representation extracted by different experts. The computation costs of soft MoEs would not be saved because all of the experts had to be executed. Our DyCC-Net does not follow the soft MoEs paradigm but instead adopts its hard version by selectively executing experts, conditioned on inputs, for more efficiency.

In addition to their architecture design, it is essential to train DyNNs [30]. DyNNs usually include non-differentiable decision functions. These functions make discrete decisions to adapt their architectures based on inputs. This poses a new challenge for training DyNNs because parameter gradients cannot be calculated via back-propagation [31]. Reinforcement learning (RL) has been utilized for the optimization of non-differentiable decision functions. In the RL paradigm, agents' parameters are optimized to make discrete decisions for dynamic inference [27,32,33]. However, the training of RL can be expensive since it is a multi-stage optimization procedure, involving training backbone networks and optimizing

the decision module. Instead of RL, our DyCC-Net uses a reparameterization technique, namely the Gumbel–Softmax function, to train DyNNs for non-differentiable decisions.

*2.2. Drone-View Object Detection*

The research of drone-view object detection is a popular area of research in the field of remote sensing. Different from natural images, drone-captured images usually contain numerous small objects. Having numerous small objects crammed together in drone-captured images is the main cause of the drop in detection performance. In this section, we summarize three major streams of drone-view object detection. SR-based drone-view detectors [11,12,17,34–37] were introduced to utilize super-resolution techniques to restore low-quality RoIs into high-quality ones. These models usually contain proposals for crowded regions, RoI super-resolution, and the final object detection modules. They are inefficient and also end-to-end learning can be hard to train. Context-based drone-view detectors [8,9,38–40] built the relationships between objects and their surrounding environments into their original RoIs' features. It is difficult to build such context relationships because of the diversity and complexity of the backgrounds in drone-captured images. Leveraging contextual information can also introduce much background noise, resulting in poor performance in detection. MR-based drone-view detectors [41–43] combined spatial and semantic information from low-level and high-level representations, respectively, for fine-grained object classification.

These drone-view detectors usually use complicated modules for super-resolution or context collection of RoIs to obtain maximum accuracy, leading to high computational costs. They neglect inputs of different levels of complexities. Our DyCC-Net focuses on the design of a drone-view detector for input-aware inference, to obtain a balanced trade-off between performance and computational cost.

## 3. Preliminaries

In this work, we adopt the Feature Pyramid Network (FPN) [44] to extract multi-scale representations and use Context Collector [19] to collect contextual information. In this section, for the integrity of this paper, we briefly introduce FPN and Context Collector as background knowledge.

*3.1. Feature Pyramid Network*

FPN [44] can effectively build multi-scale representations from different layers in a backbone and improve the network's performance. Figure 2 shows that FPN utilizes the top–down pathway and lateral connections (shown with red arrows) to aggregate the detailed spatial representations from low-level layers and the rich semantic representations from high-level layers.

Mathematically, let $R_i$ be the $i$-th top-down layer of the FPN and $F_i$ be the $i$-th lateral connection of the FPN. FPN outputs a set of feature maps $\{P_i | i = 1, 2, \ldots, S\}$, where $S$ is the number of FPN stages. The output $P_i$ is defined as:

$$P_i = R_i(P_{i+1}) + F_i(C_i), i = 1, 2, \ldots, S - 1, \tag{1}$$

where $\{C_i | i = 1, 2, \ldots, S\}$ are the inputs of FPN, and $P_S = F_S(C_S)$.

The lateral connection $F_i(C_i)$ is employed to increase or reduce the number of feature channels for the subsequent operation of feature concatenation in Equation (1). However, the features adjusted by $F_i(C_i)$ generally lack contextual information, especially for tiny objects, as the size of the convolution filter is fixed and small.

Recently, a Context Collector module [19], specially designed for drone-view detectors, was developed to improve the model's representation capability of small-size targets. The Context Collector has improved the model's performance in detecting small-size targets by collecting both local and global contextual cues. However, this is achieved at the cost of increased overall computation, especially for easy cases containing mostly large objects that can be detected without executing the context collector. Thus, in this paper, we develop

a dynamic context collector, which dynamically adapts its structure to inputs of different complexities and perform input-aware inference. Next, we briefly introduce the Context Collector before illustrating our methodology in details.

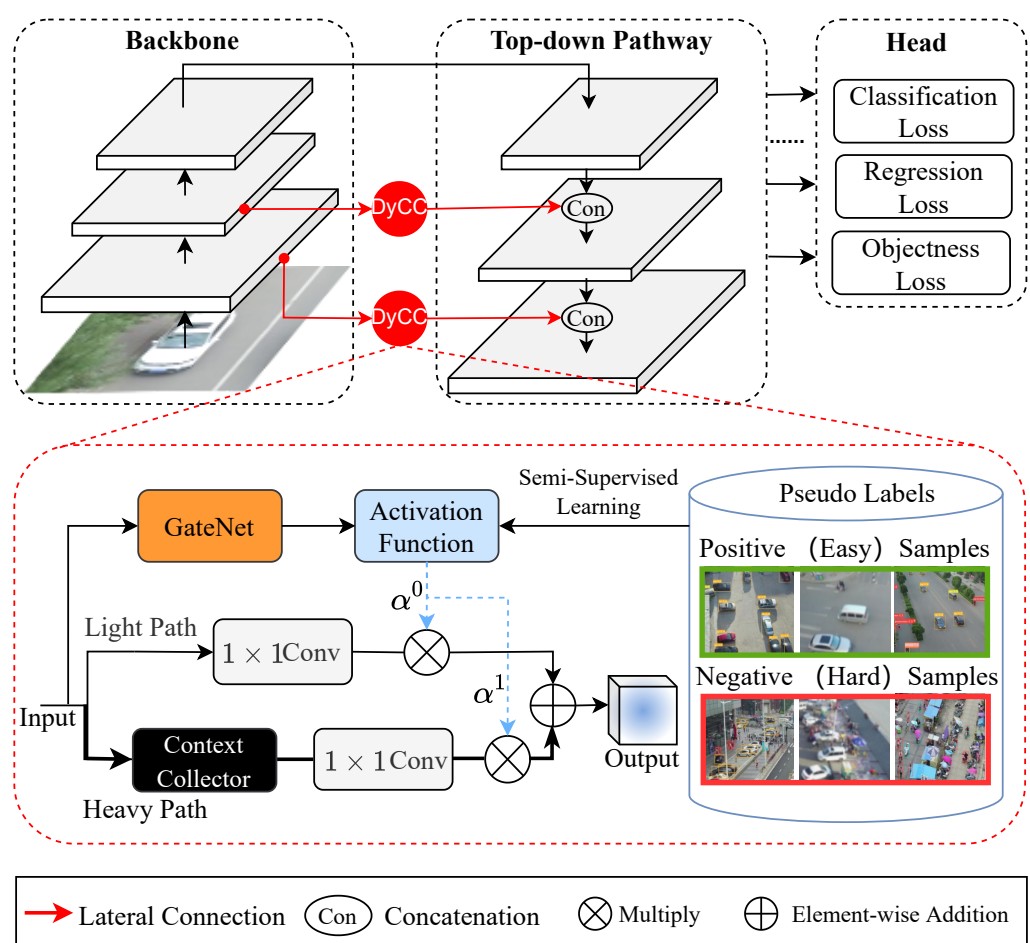

**Figure 2.** The pipeline of DyCC-Net is composed of four modules, i.e., a feature-extraction Backbone module, our input-aware DyCC module, a top–down multi-scale features-extraction pathway, and a Head for estimating bounding box positions and classification scores.

*3.2. Context Collector*

Figure 3 [19] illustrates the structure of Context Collector (CC). As it shows, CC consists of three components: a $1 \times 1$ convolutional filter, a dilated convolution for local contextual information, and a global average pooling layer for global contextual information. For the first branch, the $1 \times 1$ convolutional filter $\psi_i$ is adopted to regulate the number of the input features $C_i$. For the second branch, a few $3 \times 3$ atrous convolution filters $v_i^k$ with the dilation rate of $(1, 2, \ldots, N)$ are adopted to extract local contextual features. For the last branch, a Global Average Pooling layer $\phi_i$ is utilized to collect global context information. Then, features generated from the above branches are concatenated to obtain the final features. The above procedure can be formulated as:

$$F_i(C_i) = \sum_{k=1,2,\ldots,N} Con(v_i^k(C_i), \psi_i(C_i), \phi_i(C_i)) \tag{2}$$

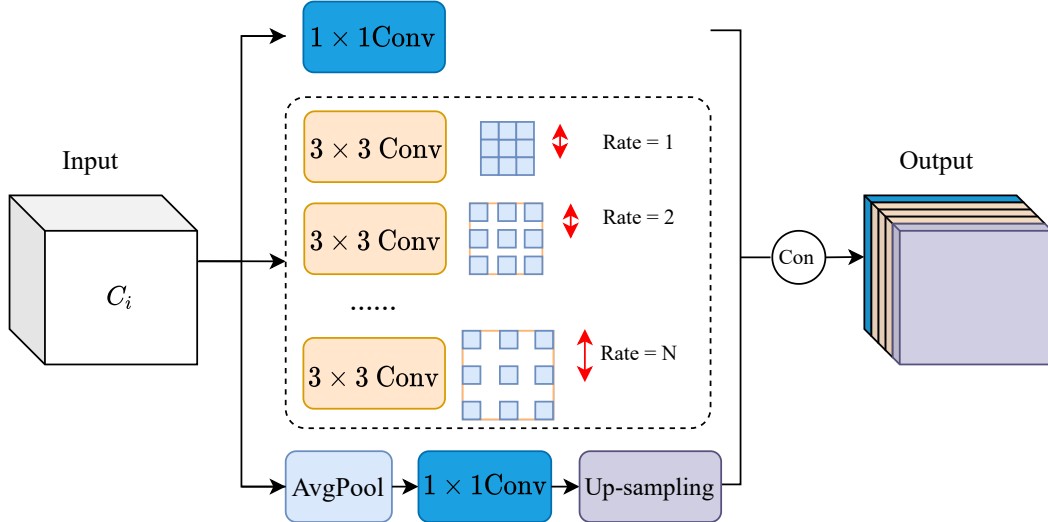

**Figure 3.** The architecture of the Context Collector [19] adopted in this work.

The Context Collector improves the expressive power of small-size targets' representations by collecting contextual information surrounding the targets. However, this is achieved as the cost of the increased overall computational costs, not desirable for drone-view detectors. Moreover, for some UAV images, the operation of context collection is not necessarily required to detect objects in the images. For example, when a flying drone is close to the ground the objects appear to be relatively large in the image captured by the drone. The network architecture of the detectors needs to be adaptive to the inputs, dynamically. Inspired by the success on DyNNs, we propose to integrate the dynamic mechanism into CC and design a dynamic context collector to achieve a better efficiency-accuracy trade-off.

## 4. Methodology

### 4.1. Overview

We first give the overview of the pipeline of the proposed DyCC-Net in Figure 2. The extra core module of our DyCC-Net is the dynamic context collector module "DyCC", in addition to the commonly used Backbone for extracting features, the standard Head for predicting the bounding box position and classification score, and a top–down pathway to obtain multi-scale representations.

Intuitively, some easy images, mainly containing large objects, can be recognized correctly without requiring rich contextual information. Therefore, a static design equipped with a contextual information collector contains computation redundancy. To alleviate such redundancy, we propose a dynamic architecture, **D**ynamic **C**ontext **C**ollector (DyCC), which can skip or execute the CC module to avoid unnecessary computation for easy inputs.

DyCC aims to reduce computational costs by evaluating input images and allowing easy image to skip the Context Collector module. During training, the computational cost of DyCC-Net does not decrease because both Light and Heavy paths are executed for either easy or hard images. During inference, its computational cost decreases by executing the Light path instead of the Heavy path for easy inputs. As detailed in the lower part of Figure 2, DyCC contains three components, i.e., a Dynamic Gate, a Heavy Path, and a Light Path. The Dynamic Gate (detailed next in Section 4.2) is responsible for predicting a gate signal, which determines the appropriate path for different inputs, i.e., the Heavy or Light Path. The Heavy Path uses the Context Collector [19] to perform context collection and $1 \times 1$ convolution for hard images, whereas the Light Path is a simple $1 \times 1$ convolutional layer that regulates the shapes of extracted features by a top–down pathway.

### 4.2. Dynamic Gate

The Dynamic Gate is designed to learn to produce a gate signal $\alpha \in \mathbb{R}^2$ based on the feature maps $C_i$ of the input images. The signal $\alpha$ is an approximate one-hot vector, which is approximately equal to a value of $[0, 1]$ when selecting the Heavy Path and $[1, 0]$ when selecting the Light Path. In the training stage, the two elements $\alpha^0$ and $\alpha^1$ are multiplied by the output of the two paths, respectively. In the testing stage, the Heavy Path is bypassed and the Light Path is executed if the output of the Dynamic Gate is approximately equal to $[1, 0]$. The Dynamic Gate consists of a Gating Network and a Gating Activation Function, as detailed below.

#### 4.2.1. Designs of Gating Network

The Gating Network (GateNet) is expected to not only accurately select which path to execute and but also to be computationally inexpensive. We investigated three different designs of GateNet structures, as shown in Figure 4. The first Gating Network, denoted by "GateNet-I" (as shown in Figure 4a) is formed by a Global Average Pooling layer $GAP$, two fully-connected (FC) layers $FC_1$ and $FC_2$, and a ReLU layer $\delta$ which outputs a two-dimensional vector. Mathematically, the output of GateNet-I, denoted by $\pi_i^{G_I}$, can be defined as:

$$\pi_i^{G_I} = FC_2(\delta(FC_1(GAP(C_i)))), \tag{3}$$

where $C_i$ is the input at the $i$-th layer of FPN. Let the shape of the input feature of GateNet be $H_i \times W_i \times C_i$ and the shape of its output feature be $H_o \times W_o \times C_o$, the computational cost of GateNet-I is about $\frac{1}{H_o \times W_o}$ of the Light Path. Although its computational cost is almost negligible, the features extracted by GateNet-I lack contextual information because the direct $GAP$ layer on $C_i$ utilizes a $1 \times 1$ value to represent a $H_i \times W_i$ feature map.

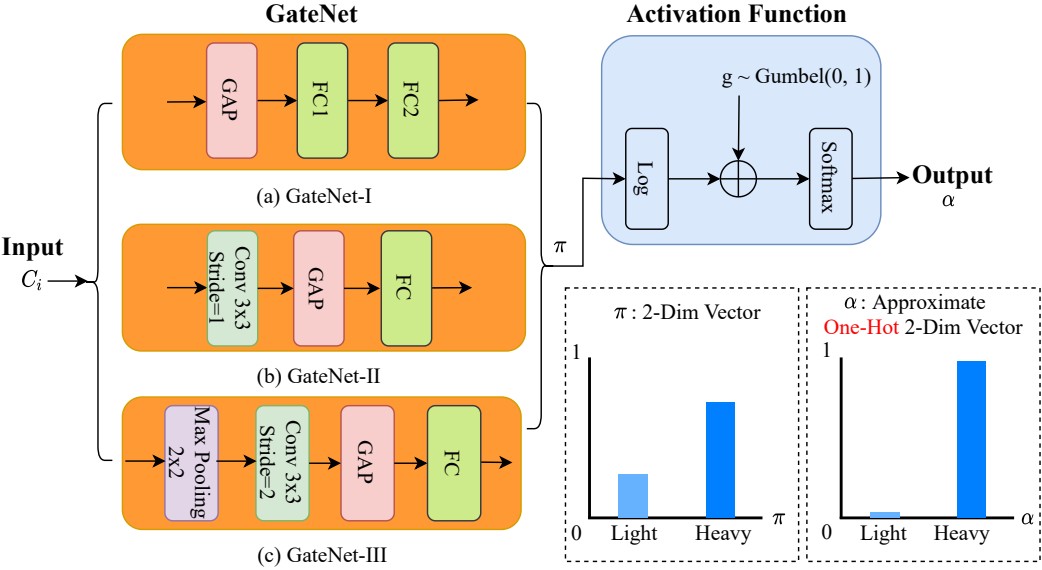

**Figure 4.** The three different architectures of Gating Network GateNet. (**a**) The computational cost of the GateNet-I is about $\frac{1}{H_o \times W_o}$ of the Light Path. (**b**) GateNet-II contains a Convolutional layer and has a computational cost about 10 times of the Light Path. (**c**) GateNet-III includes one Pooling layer and has similar computational cost to the Light Path. In addition, a Gumbel–Softmax function is utilized to convert $\pi$ to an approximate one-hot vector $\alpha$.

A convolutional layer is utilized to enrich contextual information contained in the features. Figure 4b compares the second design of the Gating Network "GateNet-II", which adopts a $3 \times 3$ convolutional layer $Conv$ for contextual information collection. Then, following the convolutional layer, a Global Average Pooling layer $GAP$ captures the context

information at image level. Finally, a fully connected layer *FC* is adopted to calculate a two-dimensional vector. Thus, the output of GateNet-II $\pi_i^{G_{II}}$ can be mathematically formulated:

$$\pi_i^{G_{II}} = FC(GAP(Conv(C_i))). \tag{4}$$

As a less computationally expensive alternative, Figure 4c presents the Gating Network (GateNet-III), which consists of a $2 \times 2$ Max pooling layer *maxP*, a $3 \times 3$ convolutional layer *Conv* using a stride of 2, *GAP* and *FC*. Similarly, the output of GateNet-III $\pi_i^{G_{III}}$ can be mathematically formulated as:

$$\pi_i^{G_{III}} = FC(GAP(Conv(MaxP(C_i)))). \tag{5}$$

The computational cost of the GateNet-III is similar to that of the Light Path. Hence, in our experiments, we use GateNet-III to determine the gate signal.

### 4.2.2. Gumbel–Softmax Gating Activation Function

An appropriate path, either Heavy or Light Path, is to be selected according to the probability distribution $\pi$ estimated by the Gating Network. The selection process is discrete and thus non-differentiable, which poses a new challenge for training DyCC-Net.

As a natural approximation, a Softmax function is widely used by existing approaches to make soft decisions in the training stage and then to revert the soft decisions to a hard version during inference. In the hard decision version, a hard threshold is required to be set during inference. While the softmax approximation method can train the DyNNs with gradients, it leads to degraded prediction accuracy ($\sim$40% drop [27]) because the network with soft decisions is not trained for the hard gating in the inference stage. The Gumbel–Softmax function [45] is adopted as the gate activation function to train the model parameters for the non-differentiable decision.

As shown in Figure 4, Gumbel–Softmax function is utilized as a continuous, differentiable function on class probabilities $\pi = \{\pi_1, \pi_2, \ldots, \pi_k\}$ and predicts a $k$-dimensional one-hot vector $\alpha$:

$$\alpha^i = \frac{\exp((\log(\pi_i) + g_i)/\tau)}{\sum_{j=1}^{k} \exp((\log(\pi_j) + g_j)/\tau)}, \tag{6}$$

where $i = 1, \ldots, k$, $g_1 \ldots g_k$ are *i.i.d.* samples drawn from Gumbel (0, 1) (The Gumbel (0, 1) distribution can be sampled using inverse transform sampling by drawing $u \sim Uniform(0, 1)$ and computing $g = -\log(-\log(u))$), and $\tau$ is a temperature parameter. The output of this activation function approximates a one-hot vector for a low $\tau$ and converges to a uniform distribution as $\tau$ increases. The Gumbel–Softmax function is a partial derivative function for the continuous distribution $\alpha$. The re-parameterization trick enables gradients to flow from $f(\alpha)$ to $\theta$ during backward propagation.

### 4.3. Pseudo Learning

The loss functions of DyCC-Net includes a classification loss for estimating objects' categories and a regression loss for estimating objects' positions. In particular, the binary cross entropy (BCE) loss is utilized for classification, which can be expressed as:

$$\mathcal{L}_{BCE} = -(y \cdot \log(\hat{y}) + (1 - y) \times \log(1 - \hat{y})), \tag{7}$$

where $y$ is the label of a sample, and $\hat{y}$ is the predicted probability of the sample. *GIoU* [46] is utilized for the regression of bounding boxes to address the weakness of *IoU* that its value is equal to zero for non-overlapping case. *GIoU* can be expressed as followed.

$$GIoU = IoU - \frac{|C - (A \cup B)|}{|C|} \tag{8}$$

$$IoU = \frac{|A \cap B|}{|A \cup B|} \tag{9}$$

where *A* is a position of a predicted bounding box, and *B* is a position of a ground truth bounding box, and *C* is the smallest convex set of $|A \cup B|$. If only the detection loss is utilized to optimize DyCC-Net, DyCC would be encouraged to take the Heavy Path as much as possible, which better minimizes the detection loss, and fail to perform input-aware inference to reduce the overall computational costs.

To address this issue, we propose a **Pseudo**-label-based semi-supervised **Learning** strategy for path selection. The training data used for the semi-supervised learning consist of automatically pseudo-labelled positive and negative image samples. Positive image samples refer to images, in which objects can be easily recognized by existing detectors, and, hence, "easy". Negative image samples refer to images in which objects are hard to recognize with existing detectors and, hence, "hard".

Figure 5 illustrates the process of the pseudo label generation. We firstly run a well-trained baseline object detection model on all images and then evaluate each image's detection result against the median detection precision of the whole set. In this work, $AP_{50}$ is adopted to evaluate the prediction precision of each image. Images whose $AP_{50}$ scores are higher than the median $AP_{50}$ score of the whole set are labeled as positive image samples, also known as easy images, and images whose $AP_{50}$ scores are lower than the median $AP_{50}$ score are labeled as negative image samples, also known as hard images. Thus, we can generate pseudo labels for all images, which are used to train the DyCC-Net to determine whether to take the Light Path or the Heavy Path given an input image.

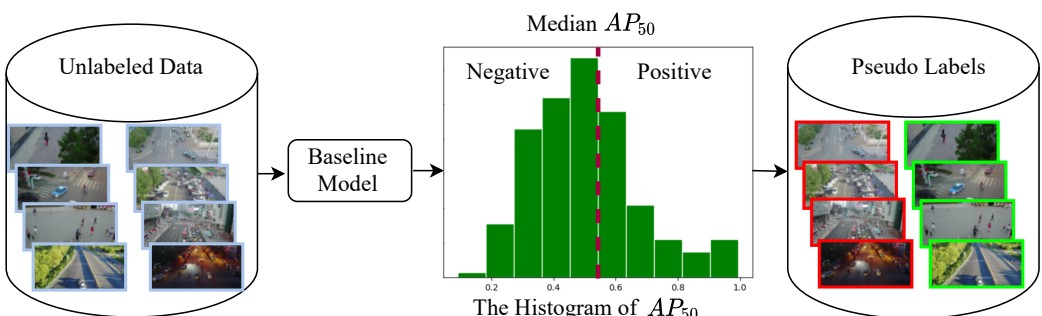

**Figure 5.** The pipeline of the pseudo label generation. Images with gray rectangles are fed into a pre-trained object detector. Images with an estimated precision higher than the threshold are treated as positive samples (shown with green rectangles) and images with an estimated precision lower than the threshold are treated as negative ones (shown with red rectangles).

Some examples of positive and negative image samples are presented in Section 5.3.4. Compared with unsupervised learning, our GateNet trained with Pseudo Learning can make a more accurate selection of the appropriate path, to obtain a better efficiency–accuracy trade-off.

Additionally, note that DyCC-Net is trained with both the generated pseudo-labels ("easy" or "hard") and the original object detection annotations (bounding boxes and object classes) of the images to achieve high-accuracy and high-efficiency object detection from drone-view images.

## 5. Experiments

The proposed DyCC-Net is evaluated on two benchmark datasets and compared with 10 SOTA methods to demonstrate that it can effectively detect objects in images captured by drones. In this section, we first present the two datasets and models, and then verify the effectiveness of DyCC-Net on the datasets for drone-view detection.

### 5.1. Datasets and Models

The recent and well-known survey [47] for drone vision reported that there are only two public datasets specially created for drone-view object detection, i.e., VisDrone2021 [47] and UAVDT [48]. Thus, we choose those two datasets to verify that our DyCC-Net can effectively detect objects in drone-captured images, as did the SOTA drone-view detectors, e.g., UFPMP-Det [18] and CRENET [49]:

(1) *VisDrone2021* [47]: The VisDrone2021 dataset contains ten object categories, e.g., pedestrian, person, car, etc. Every image in the dataset has annotations of object class and bounding box and has a resolution about $2000 \times 1500$. The VisDrone2021 is split into three subsets: 6471 images for training, 548 images for validation, and 3190 for testing.

(2) *UAVDT* [48]: The UAVDT dataset contains three object categories including bus, truck, and car. Each image in the dataset also has annotations of object class and bounding box and has a resolution of $1080 \times 540$. The UAVDT is split into two subsets: 23,258 images in the training subset, and 15,069 images in the testing subset.

We have explored several models widely-used for object detection, namely FRCNN [50], SSD [51] and adopted YOLOv5 [52] as the baseline model for the overall performance comparison. In addition, our method is compared with SOTA specially designed for drone-view detection, i.e., UFPMP-Det [18], CRENET [49], TPH-YOLOv5 [53], DSHNet [54], CRENET [49], GLSAN [12], and ClustDet [17].

### 5.2. Implementation and Evaluation Metrics

(1) *Implementation*: All of the experiments are conducted using one NVIDIA RTX3090 GPU; DyCC-Net is implemented with PyTorch 1.8.1. During training, the pre-trained model YOLOv5 [52] is used as the backbone. The Stochastic Gradient Descent (SGD) optimizer is used for training DyCC-Net and the learning rate with a Cosine learning rate schedule is initialized to $6 \times 10^{-5}$. The long side of the input images is 1536 pixels, as did TPH-YOLOv5 [53].

(2) *Evaluation Metrics*: The detection performance of the proposed DyCC-Net is evaluated using the same metrics as PASCAL VOC [55], i.e., mean Average Precision (*mAP*) and Average Precision (*AP*), which are defined by:

$$mAP = \frac{1}{M} \sum_{i=1}^{M} AP_i,$$ (10)

where $M$ is the number of objects' categories, and

$$AP = \int_0^1 P(R)dR.$$ (11)

Here, $R$ is Recall, measuring how good the classifier estimates the positives and calculated as the percentage of true positive predictions in the total number of positive samples, $P$ is Precision, measuring how accurate the prediction is and calculated as the percentage of correct positive predictions in the total number of positive predictions, and $P(R)$ is the precision-recall curve. $P$ and $R$ are defined as follows:

$$P = \frac{TP}{TP + FP}$$ (12)

and

$$R = \frac{TP}{TP + FN}$$ (13)

where $FN$, $FP$ and $TP$ indicate the numbers of false negatives, false positives, and true positives, respectively. $AP$ is averaged on 10 Intersection over Union (IoU) values of [0.50: 0.05: 0.95], where $AP_{50}$ is calculated at the single IoU of 0.5.

### 5.3. Ablation Studies

We conduct extensive ablation studies on VisDrone2021 in order to verify the contributions of each component in DyCC-Net. YOLOv5 [52] is utilized as the baseline.

#### 5.3.1. The Effectiveness of CC

We first demonstrate the effectiveness of CC to show that detection performance can be boosted by considering context information with the context collecting module. We show the baseline's detection performance with the configuration of dilation rate $d = 1$ and kernel size $k = 1$. Firstly, a $3 \times 3$ convolution layer in each lateral connection is added, leading to 0.7 gain in $AP_{50}$. Next, as shown in Table 1, as we gradually add more convolution filters, the model's performance continues improving. The configuration in the last row benefits from all convolution filters and, thus, obtains the best performance of 41.1% and improves the baseline by 1.9% in $AP_{50}$. Thence, the convolution filter of the CC finally adopts the following configuration: dilation rate $d = [1, 1, 2, 3, 4, 5]$ and kernel size $k = [1, 3, 3, 3, 3, 3]$. We denote the CC with such a configuration as $CC_{k6d6}$.

**Table 1.** The effectiveness of CC under different configurations of dilation rate $k$ and kernel size $k$.

| k = 1<br>d = 1 | k = 3<br>d = 1 | k = 3<br>d = 2 | k = 3<br>d = 3 | k = 3<br>d = 4 | k = 3<br>d = 5 | $AP_{50}[\%]$ |
|---|---|---|---|---|---|---|
| ✓ | | | | | | 39.2 |
| ✓ | ✓ | | | | | 39.9 |
| ✓ | ✓ | ✓ | | | | 40.4 |
| ✓ | ✓ | ✓ | ✓ | | | 40.8 |
| ✓ | ✓ | ✓ | ✓ | ✓ | | 41.0 |
| ✓ | ✓ | ✓ | ✓ | ✓ | ✓ | 41.1 |

#### 5.3.2. The Effectiveness of DyCC

In this section, we aim to demonstrate that the proposed DyCC can reduce computation costs on the whole testing set while preserving detection accuracy. The computation costs are measured as the average floating point operations (FLOPs) required to process each image in the dataset. We have tested three configurations of YOLOv5 models, i.e., small YOLOv5 (denoted as "YOLOv5-s"), medium YOLOv5 (denoted as "YOLOv5-l"), and large YOLOv5 (denoted as "YOLOv5-x"). Moreover, the input image size is $640 \times 640$ pixels.

Table 2 quantitatively compares the computation costs and detection accuracy with or without the proposed DyCC. In this table, YOLOv5+CC and YOLOv5+DyCC means the baseline model integrated with $CC_{k6d6}$ and DyCC, respectively. The figures in the last row of Table 2 show that the computation costs of YOLOv5-s, YOLOv5-l, and YOLOv5-x with the proposed DyCC are 11.31G, 72.12G, and 134.47G with 34.39%, 43.47%, and 44.87% of $AP_{50}$, respectively. Compared with their baseline results in the first row of the table, adopting our proposed DyCC improves the detection accuracy of the baseline by about 1% with only a slightly increased computation costs. Compared with their counterparts shown in the second row of the table, YOLOv5 with DyCC significantly reduces the computation costs of YOLOv5+CC by about 10%, while retaining similar detection accuracy. This is because the proposed DyCC processes easy images with Light Path and only hard images with Heavy Path, thus reducing the overall computation costs on the whole testing set and retaining the detection accuracy. Additionally, note that the computation costs of YOLOv5 models are about half of those reported in [52] because the type of tensors we used is "torch.HalfTensor" instead of "torch.FloatTensor".

Table 3 quantitatively compares the three proposed GateNets. The first row of Table 3 presents that the computation costs of the proposed GateNet-I integrated into YOLOv5-s, YOLOv5-l and YOLOv5-x are 0.30G, 1.19G, and 1.87G with 34.36%, 43.45% and 44.86% of $AP_{50}$, respectively. The second row of Table 3 presents GateNet-II can significantly improve the computation costs by 5~10 times because of the newly added convolutional

layer in GateNet. The last row of Table 3 presents GateNet-III improves the detection performance with a slight improvement of the computation costs because of the newly added convolutional and max pooling layer. Hence, GateNet-III is chosen in DyCC for path selection.

**Table 2.** Effectiveness of the proposed DyCC and Pseudo Learning (indicated as "PL") on computation cost (measured by "FLOPs (G)") and detection accuracy (measured by "$AP_{50}$"). The text 's', 'l' and 'x' refer to small YOLOv5 ("YOLOv5-s"), medium YOLOv5 ("YOLOv5-l"), and large YOLOv5 ("YOLOv5-x"), respectively.

| Method | FLOPs (G) | | | $AP_{50}$(%) | | |
|---|---|---|---|---|---|---|
| | s | l | x | s | l | x |
| YOLOv5 | 10.09 | 68.28 | 129.03 | 33.50 | 42.40 | 43.94 |
| YOLOv5 + CC | 13.15 | 79.41 | 146.08 | 34.43 | 43.51 | 44.89 |
| YOLOv5 + DyCC w/o PL | 13.50 | 80.87 | 148.36 | 34.41 | 43.50 | 44.89 |
| **YOLOv5 + DyCC (w PL)** | **11.31** | **72.12** | **134.47** | **34.39** | **43.47** | **44.87** |

**Table 3.** Comparison of different GateNets in term of computation cost (measured by "FLOPs (G)") and detection accuracy (measured by "$AP_{50}$").

| GateNet | FLOPs (G) | | | $AP_{50}$(%) | | |
|---|---|---|---|---|---|---|
| | s | l | x | s | l | x |
| GateNet-I | 0.30 | 1.19 | 1.87 | 34.36 | 43.45 | 44.86 |
| GateNet-II | 1.50 | 5.96 | 9.31 | 34.42 | 43.51 | 44.90 |
| **GateNet-III** | **0.38** | **1.49** | **2.33** | **34.39** | **43.47** | **44.87** |

5.3.3. The Effectiveness of Pseudo Learning

In this section, we show that the proposed Pseudo Learning strategy can guide our Dynamic Gate to distinguish between easy and hard inputs, so that different inputs take different paths to avoid unnecessary computation. The effectiveness of Pseudo Learning is evaluated as its impact on the overall performance and efficiency. We also visually show some examples of the positive and negative image samples classified by the Dynamic Gate.

Table 2 also quantitatively compares the computation costs (measured by "FLOPs") and detection accuracy (measured by "$AP_{50}$") obtained with or without using the proposed Pseudo Learning. Figures in the last row of the table show that the FLOPs of YOLOv5-s, YOLOv5-l and YOLOv5-x with using Pseudo Learning (indicated as "w PL") are 11.31 G, 72.12 G, and 143.47 G with $AP_{50}$ of 34.39%, 43.47%, and 44.87%, respectively. Compared with their counterparts in the third row of the table (indicated as "w/o PL"), adopting our proposed Pseudo Learning reduces the computation costs by about 10%, while retaining the detection accuracy. This is because without the guidance of the proposed Pseudo Learning, DyCC tends to take the Heavy Path as much as possible even for very easy images to minimize the detection loss, which results in unnecessary computation.

Figure 6 shows image examples classified as positive and negatives by our Dynamic Gate. As can be seen that, images in the same group share similar characteristics in terms of the objects' size and density. In general, objects in positive image samples are larger and distributed with a lower density, whereas objects in the "Negative" examples are smaller and more crowded. Our Dynamic Gate can identify the object difference in the input images and select the appropriate path accordingly. We have achieved a classification accuracy 80.95%, which shows that our Dynamic Gate can correctly distinguish positive and negative image samples.

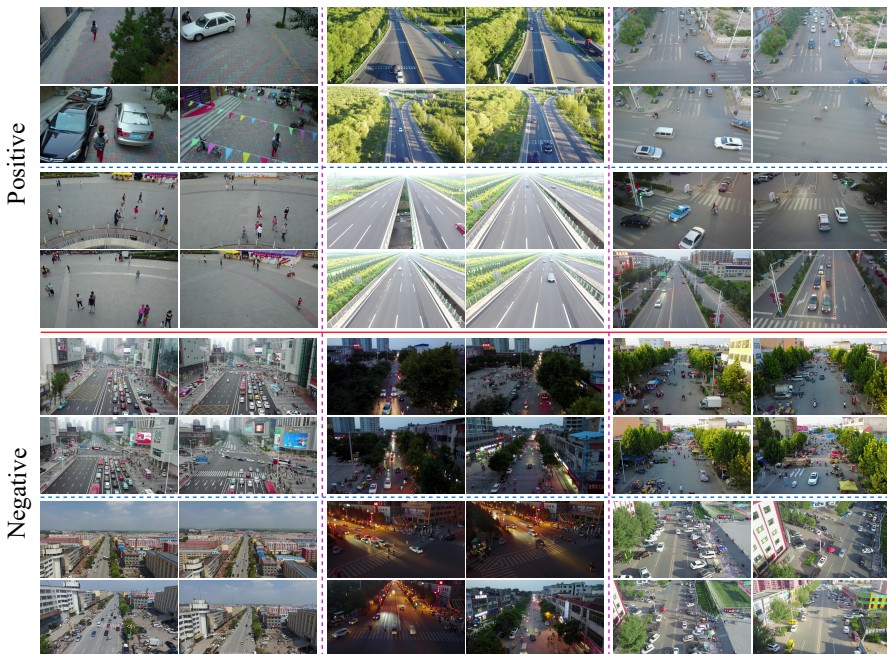

**Figure 6.** Visualization of positive and negative image samples classified by our Dynamic Gate in the DyCC-Net. Positive image samples in the top two rows are processed with the Light Path and negative image samples in the bottom two rows are processed with the Heavy Path.

### 5.3.4. Generation of Pseudo Labels

In Section 4.3, we provide details of the generation of pseudo labels, where images whose $AP_{50}$ scores are higher than the median $AP_{50}$ score of the whole set are labeled as positive image samples and images whose $AP_{50}$ scores are lower than the median $AP_{50}$ score are labeled as negative image samples. In this subsection, we show the median $AP_{50}$ scores collected from the training set and validation set, and some examples of pseudo-labeled positive and negative images with their $AP_{50}$ scores.

Figure 7a,b show the histograms of $AP_{50}$ of images in the training and validation sets, respectively. For the training set, the median $AP_{50}$ score of 0.73 is obtained and used to distinguish between positive and negative samples. Thus, images whose $AP_{50}$ scores are higher than 0.73 are labeled as positive image samples, also known as easy images, whereas images whose $AP_{50}$ scores are lower than 0.73 are labeled as negative image samples, also known as hard images. Similarly, the median $AP_{50}$ score of 0.49 is obtained and used to create the pseudo labels for images in the validation set. Note that, the median $AP_{50}$ score 0.73 obtained from the training set is higher than the median $AP_{50}$ score 0.49 of the validation set. This can be explained as the model trained with the training images fits the training set better than the validation images.

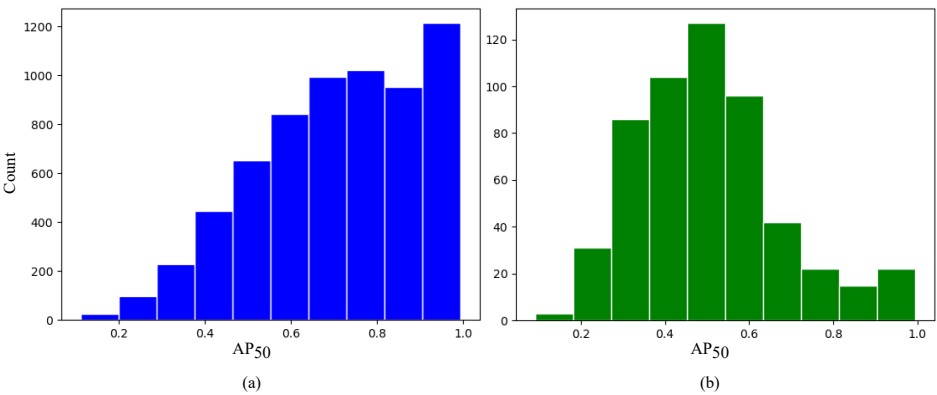

**Figure 7.** The histograms of $AP_{50}$ of images in the training set (**a**) and validation set (**b**).

Figure 8 shows some more examples of pseudo-labeled positive and negative images. As it can be seen, targets in the images labeled as "Positive" (in the first row) can be easily recognized by the baseline object detector with high $AP_{50}$ scores. In contrast, targets in the images labeled as "Negative" (in the second row) are hard to recognize with the baseline detector. When zoomed in to those missing-detection areas (shown in green rectangles), lots of small objects, including pedestrians and motors, have been missed by the baseline object detector.

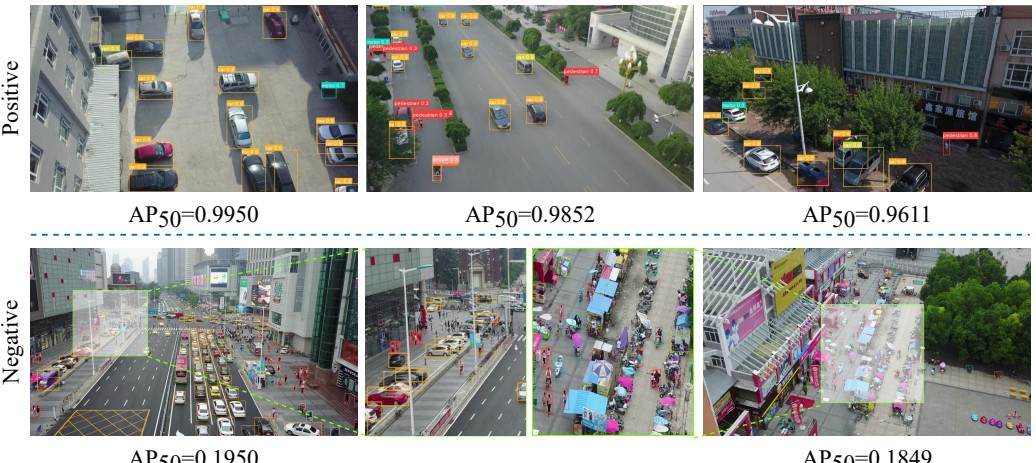

AP$_{50}$=0.9950    AP$_{50}$=0.9852    AP$_{50}$=0.9611

AP$_{50}$=0.1950    AP$_{50}$=0.1849

**Figure 8.** Examples of positive and negative image samples obtained by applying the baseline model. Positive image samples contain few, sparsely distributed, large sized objects in the first row. Negative image samples contain large number of objects, and crowded and small objects.

5.3.5. Analysis of Performance Gain and Complexity of DyCC-Net

The performance gain is brought by multiple factors. In this section, we dive in and unveil the detailed performance gains contributed by each factor. This is shown in Table 4. In this table, DyCC-Net w/o DyCC (YOLOv5+tinyHead+CC) means that a prediction head [53] for low-level feature maps and CC are integrated into YOLOv5. As shown in the the last row of the table, that our DyCC-Net can reduce the computation costs of DyCC-Net w/o DyCC by about 10% (FLOPs of 456.17G vs. 505.46G), while achieving similar detection performance (a Recall of 57.01% vs. 57.16% and an $AP_{50}$ of 59.72% vs. 59.98%). Moreover, the size of input images also affects the detection accuracy of drone-view object detection because drone-captured images typically contain a large number of small objects, which may become detectable when the image resolution becomes higher.

We also compare the training time of our DyCC-Net with the baselines. For fair comparisons, all models were trained with a batch size of four on one NVIDIA RTX3090 GPU. The training time of a model is the time interval between the start of training and when the model converges and achieves the detection accuracy $AP_{50}$ in Table 4. The last column in Table 4 shows that the training time gradually increases with a few factors, i.e., input image size, extra modules, and our DyCC module. However, in Section 5.4 (3), our approach has significantly decreased the computation cost and inference time, which is critical for the practical application of such embedded systems.

**Table 4.** The analysis of detection accuracy of the proposed DyCC-Net.

| Method | Image Size | Recall [%] | $AP_{50}$ [%] | FLOPs (G) | Training Time (h) |
|---|---|---|---|---|---|
| YOLOv5 | $640 \times 640$ | 41.59 | 42.40 | 68.28 | 8.3 |
| YOLOv5 | $1556 \times 1556$ | 53.76 | 55.60 | 392.89 | 23.0 |
| YOLOv5 + tinyHead | $1556 \times 1556$ | 56.34 | 58.59 | 440.08 | 32.7 |
| DyCC-Net w/o DyCC | $1556 \times 1556$ | 57.17 | 59.98 | 505.46 | 60.0 |
| DyCC-Net | $1556 \times 1556$ | 57.01 | 59.72 | 456.17 | 91.7 |

### 5.4. Comparison with SOTA Models

We now compare DyCC-Net with the SOTA models on VisDrone2021 and UAVDT in Table 5. We present $AP[\%]$, $AP_{50}[\%]$, $AP_{75}[\%]$ and inference time reported in their original papers. The results clearly demonstrate that DyCC-net obtains the best balance between time efficiency and detection performance.

(1) *Results on VisDrone2021:* Table 5 compares the detection results of some detectors on VisDrone2021, including one-stage detectors SSD [51] and YOLOv5 [52], and two-stage detectors FPN [44] and FRCNN [50]. DyCC-Net achieves an $AP$ of 40.07%, $AP_{50}$ of 59.72%, and $AP_{75}$ of 42.14%, which outperforms the previous detectors. The performance comparison with the SOTA detectors specially designed for aerial images, namely UFPMP-Det [18], TPH-YOLOv5 [53], DSHNet [54], CRENet [49], GLSAN [12], and ClustDet [17], is also presented in Table 5. DyCC-Net outperforms UFPMP-Det [18] by large margins of 1.94% in $AP_{75}$ and 0.87% in $AP$. Figure 9 shows the detection results on aerial images. Please note, we do not utilize tricks, e.g., model ensembles or oversized backbones, which are usually adopted in existing models for drone-captured images.

(2) Results on UAVDT: Table 5 also presents the performance comparison of DyCC-Net and SOTAs on the UAVDT dataset, i.e., UFPMP-Det [18], ClusDet [17], FRCNN [50], GLSAN [12], and YOLOv5 [52]. DyCC-Net achieves an $AP$ of 26.91%, $AP_{50}$ of 39.63% and $AP_{75}$ of 31.44%, which outperforms UFPMP-Det [18] by large margins of 2.31% in $AP$, 0.93% in $AP_{50}$, and 3.44% in $AP_{75}$.

(3) Overall Complexity: We show the inference time cost, in comparison to ClusDet [17], CRENet [49], and UFPMP-Det [18], TPH-YOLOv5 [53] to evaluate the time efficiency of DyCC-Net. All the models are evaluated using a GTX 1080Ti GPU, except for CRENet [49] on a RTX 2080Ti GPU. Table 6 shows that, DyCC-Net reduces redundant computation by input-aware inference and, thus, achieves a significantly faster inference speed. Moreover, UFPMP-Det performs inference in a coarse-to-fine fashion, where a coarse detector is used to find sub-regions containing small and densely distributed objects, and then a fine detector is adopted to these areas to locate small targets. To obtain detection performance comparable to DyCC-Net, UFPMP-Det has to spend more time.

**Table 5.** Comparison of DyCC-Net with the SOTA models on UAVDT and VisDrone2021 datasets. '-' denotes that the corresponding experimental statistics are not available. We highlight the top two results in red and green.

| Method | Reference | VisDrone2021 | | | UAVDT | | |
|---|---|---|---|---|---|---|---|
| | | $AP(\%)$ | $AP_{50}(\%)$ | $AP_{75}(\%)$ | $AP(\%)$ | $AP_{50}(\%)$ | $AP_{75}(\%)$ |
| SSD [51] | ECCV16 | - | 15.20 | - | 9.30 | 21.40 | 6.70 |
| FRCNN [50] + FPN [44] | CVPR17 | 21.80 | 41.80 | 20.10 | 11.00 | 23.40 | 8.40 |
| YOLOv5 [52] | Github21 | 24.90 | 42.40 | 25.10 | 19.10 | 33.90 | 19.60 |
| DSHNet [54] | WACV21 | 30.30 | 51.80 | 30.90 | 17.80 | 30.40 | 19.70 |
| GLSAN [12] | TIP20 | 30.70 | 55.60 | 29.90 | 19.00 | 30.50 | 21.70 |
| ClustDet [17] | ICCV19 | 32.40 | 56.20 | 31.60 | 13.70 | 26.50 | 12.50 |
| CRENet [49] | ECCV20 | 33.70 | 54.30 | 33.50 | - | - | - |
| TPH-YOLOv5 [53] | ICCVW21 | 35.74 | 57.31 | - | - | - | - |
| mSODANet [56] | PR22 | 36.89 | 55.92 | 37.41 | - | - | - |
| UFPMP-Det [18] | AAAI22 | 39.20 | 65.30 | 40.20 | 24.60 | 38.70 | 28.00 |
| **DyCC-Net** | **Ours** | 40.07 | 59.72 | 42.14 | 26.91 | 39.63 | 31.44 |

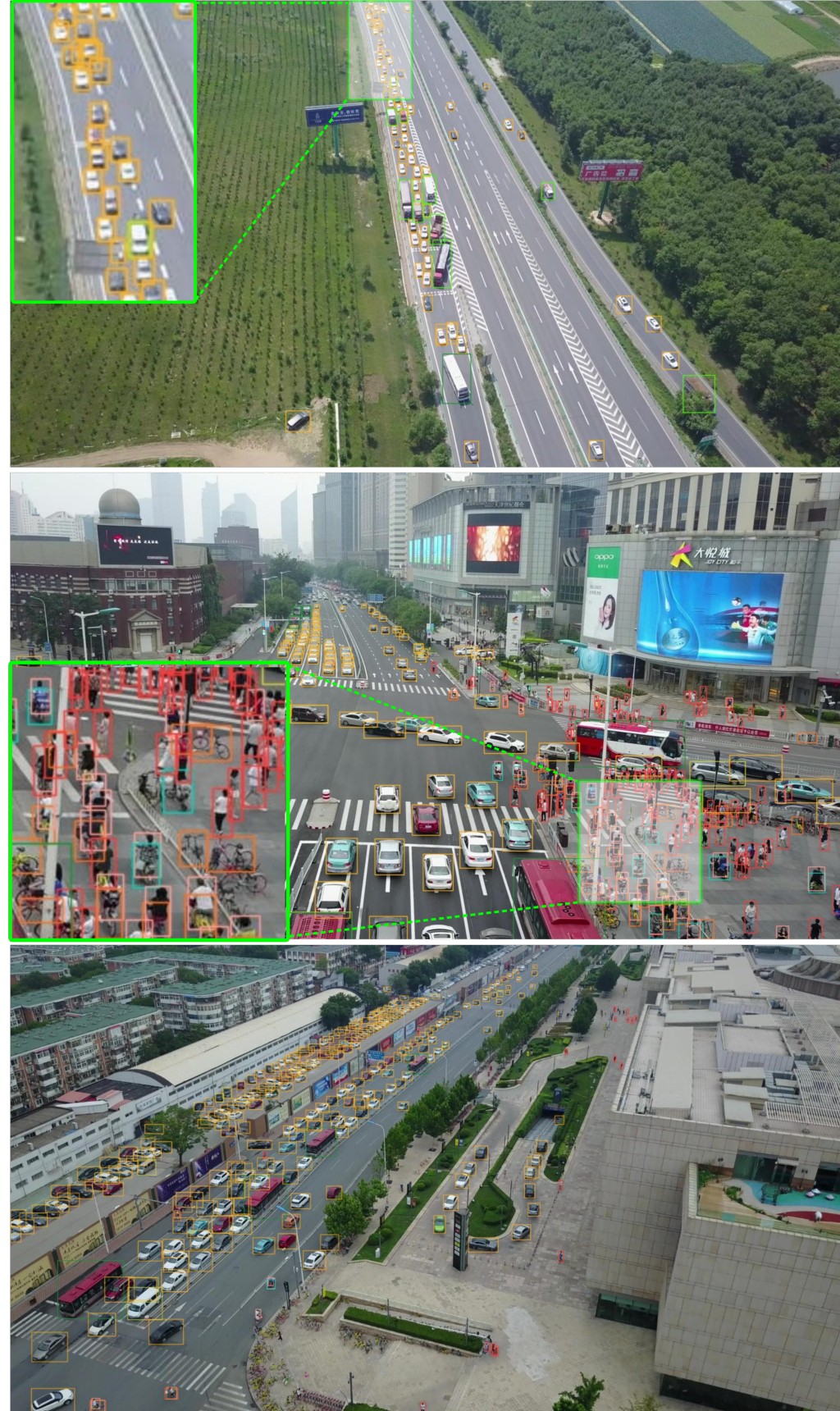

**Figure 9.** Visualization of the detection results obtained with our DyCC-Net.

**Table 6.** Comparison of different methods in inference time cost (s) on VisDrone. '-' denotes that the corresponding experimental statistics are not available. '*' means that authors do not release their source codes.

| Method | Inference Time (s) | Params (M) | FLOPs (G) |
|---|---|---|---|
| ClusDet [17] | 0.273 | - | - |
| TPH-YOLOv5 [53] | 0.305 | 60.45 | 498.85 |
| CRENet [49] * | 0.901 | - | - |
| UFPMP-Det [18] * | 0.152 | - | - |
| **DyCC-Net (Ours)** | **0.105** | **46.11** | **456.17** |

## 6. Conclusions

In this paper, we have proposed DyCC-Net, which can perform input-aware inference for effective UAV object detection by dynamically adapting its structure to the input image. Our DyCC-Net can reduce the computational cost by input-aware inference without sacrificing prediction accuracy. To address the non-differentiability of path selection, we have introduced Gumbel–Softmax to perform gradient backpropagation during training. Moreover, we have proposed Pseudo Learning to make a more robust and accurate selection of paths based on diverse inputs.

We have evaluated DyCC-Net on two drone-captured datasets and compared DyCC-Net with 10 SOTA models. Experiment results have demonstrated that the proposed DyCC-Net has achieved high time efficiency while preserving the original accuracy. Compared with the SOTA drone-view detectors, the proposed DyCC-Net has achieved comparable accuracy with less inference time costs. Moreover, extensive ablation studies have further demonstrated the effectiveness of each module of DyCC-Net. Our DyCC-Net offers the potential to reduce computational cost and can be efficiently deployed on drones. Finally, to further address the issue that large inputs bring large computation cost, we plan to investigate spatial-wise DyNNs to explore "spatially dynamic" computation to further reduce the computational cost by processing a fraction of pixels or regions in an image.

**Author Contributions:** Conceptualization, Y.X. and W.J.; Methodology, Y.X. and W.J.; Validation, Y.X.; Formal analysis, Y.X.; Writing—original draft preparation, Y.X.; Writing—review and editing, W.J., X.L., X.F. and J.L.; Visualization, Y.X.; Supervision, Q.M.; Funding acquisition, X.L. All authors have read and agreed to the published version of the manuscript.

**Funding:** The work was supported by the Key R&D Projects of Qingdao Science and Technology Plan (No. 21-1-2-18-xx). It was funded by the Fundamental Research Funds for the Central Universities (No. 20101216855).

**Data Availability Statement:** Not applicable.

**Acknowledgments:** Sincere thanks to the authors of Yolov5 and Faster RCNN for providing their algorithm codes, which are convenient for comparison experiments.

**Conflicts of Interest:** The authors declare no conflicts of interest.

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
