# Peer review of "DyCC-Net: Dynamic Context Collection Network for Input-Aware Drone-View Object Detection"

_remotesensing, doi:10.3390/rs14246313_

Round 1

Reviewer 1 Report

Comments

This manuscript has carried out some additional processing on the existing neural network framework. It improves inference efficiency by skipping or executing a context collector conditioned on the complexity of the input images.

In general, the manuscript is well structured and the experiments are abundant. It mainly proposes a Dycc module, in which GateNet and Context collector are used. In addition, Pseudo Labels are considered. I suggest a Major revision of the manuscript to further improve its quality. Here are some comments.

(1)    In lines 92-95: There are three main methods: dynamic depth, dynamic width, and dynamic routing. What are their advantages and disadvantages?Do you have an experiment comparing these three methods? Why do you finally choose to use dynamic depth?

(2)    In lines 237-238: Please give the concrete formula of the loss function.

(3)    In lines 240-241, the manuscript say “ Hence, DyCC-Net tends to select the Heavy Path even for easy images.” What is the impact of choosing a heavy path on the calculated cost?

(4)    In lines 314-315, What is the evaluation index of good or bad visualization?

(5)    Figure 9: there are not detailed information introduced about the positive and negative samples. In the Negative samples, many objects are not labeled, e.g., many persons. Please explain why.

(6)    In Table2:Please add evaluation metrics, e.g., the comparison of params, recall, FLOPs. And how DyCC-Net improves prediction accuracy?

(7)    What about the training time of the proposed network, compared to other networks?

(8)    What about the performance of the proposed network compared to existing DyNNs?

Author Response

Thank you very much for allowing major revisions of our manuscript, with an opportunity to address the reviewers’ comments. We appreciate very much your valuable suggestions and comments. We have revised our manuscript accordingly.

We are uploading the following materials for the second-round review of our revised manuscript:

  • The item-by-item response to the comments (below) (response to reviewers),
  • A revised manuscript with all changes highlighted in blue,
  • A clean revised manuscript without highlights (PDF main document).

Reviewer 2 Report

The proposed DyCC aims to reduce computational costs by allocating appropriate computational resources for easy or hard UAV images. There are some confused problems in this article.

1. In ablation studies “The Effectiveness of CC”, the metric of reference table 1 is AP, while the metric in the corresponding description is AP50.

2. In ablation studies “Visualization of the Features Enhanced by CC”, visualization results are not enough to show its effectiveness and the detection results with metrics are needed.

3. In ablation studies “The Effectiveness of DyCC”, the comparison between CC+YOLOv5 and DyCC+YOLOv5 is not fair and cannot show the effectiveness on reducing the computional complexity.of the proposed module. The authors should make comparison between the baseline YOLOv5 and DyCC+YOLOv5.  

4. In ablation studies “The Effectiveness of Pseudo Learning”, results with metrics are needed. It is useless when the results are all visualization. Several visualization results are not enough to support the conclusion in the article

5. In Table 5, the AP of the baseline YOLOv5 is 24.90% on VisDrone2021 while the AP of the proposed DyCC-Net is 40.07%. How did you improve the AP of the baseline YOLOv5 from 24.9% AP to 40.07%? It is fabulous! The authors should introduce the improvement step by step.

Author Response

(The authors gave the same response as above.)

Reviewer 3 Report

The authors proposed a dynamic neural network for object detection on drone-view images. Overall, the manuscript is well-structured. I didn’t find any major issues on this work. However, the experiment section is a little bit weak and less exciting because 

1. The context collector is proposed by another paper with the same authors. Some experiments (table 1, figure 6) are basically the same between two papers. Is it necessary to show the same experiments? We can learn how the context collector works from the last paper, and the focus of this work should be on the dynamic module.  

2. You only have one experiment on the dynamic module, and it lacks details. For example, what does it mean by “matching similar accuracy”? how much is the “smaller fractions of inputs require large networks”? Figure 7 shows the dynamic module indeed reduces the computation cost, but I didn’t see many discussions on the module other than that. The contributions you mentioned in the introduction section are about the dynamic module and input-aware inference instead of the context collector, so I would expect to see more discussions on it. The discussions can give the readers an idea of how to extend or modify your work based on different criteria. For example, you proposed three different GateNets, it would be interesting to see if there are performance differences by using different GateNets or if the computational cost matches your theory.

3. I don’t quite understand the significance of the pseudo learning experiment. In the section 4.3, you mentioned the threshold is 0.6 for separating easy and hard images. In the ablation study, you also mentioned the threshold is 0.73. Which value did you use then? In addition, why did you need the threshold value for the validation set? I expect the threshold value is used during training, and you should use the value generated form the GateNet during the validation phase. Can you further elaborate this? 

Author Response

(The authors gave the same response as above.)

Round 2

Reviewer 2 Report

Thanks for the authors' reponse. I keep the original decision for the following resons:

(1) According to the results, the problem of computation cost is not addressed. The FLOPs of the proposed method is highter than the basline model (YOLOv5). 

(2) The module (Context Collector) that contributes AP gains are less innovative. 

(3) A large improvement of the proposed method in AP is from the big scale of input images, and the large input image requires more computation cost. 

Author Response

Dear Sir or Madam,

Thank you very much for allowing minor revisions of our manuscript, with an opportunity to address the your comments. We appreciate very much your valuable suggestions and comments. We have revised our manuscript accordingly.

We are uploading the following materials for the next review of our revised manuscript:

  • The item-by-item response to the comments (below) (response to reviewers),
  • A revised manuscript with all changes highlighted in blue,
  • A clean revised manuscript without highlights (PDF main document).

Your favorable consideration on our revision would be most appreciated. Thank you!

Best regards,

Yue Xi, on behalf of all co-authors

Reviewer 3 Report

The authors have addressed the issues in the previous review. I recommend to accept the manuscript for publication. 

Author Response

Dear Sir or Madam,

We appreciate very much your valuable suggestions and comments. We have revised our manuscript accordingly. We are uploading the following materials:

  • A revised manuscript with all changes highlighted in blue,
  • A clean revised manuscript without highlights (PDF main document).

Thank you!

Best regards,

Yue Xi, on behalf of all co-authors